# Data-driven emergence of convolutional structure in neural networks

**Alessandro Ingrosso**                          INGROSSO@ICTP.IT
*The Abdus Salam International Centre for Theoretical Physics (ICTP), Trieste, Italy*

**Sebastian Goldt**                             SGOLDT@SISSA.IT
*International School of Advanced Studies (SISSA), Trieste, Italy*

**Editors:** Sophia Sanborn, Christian Shewmake, Simone Azeglio, Arianna Di Bernardo, Nina Miolane

## Abstract

Exploiting data invariances is crucial for efficient learning in both artificial and biological neural circuits, but can neural networks learn apposite representations from scratch? Convolutional neural networks, for example, were designed to exploit translation symmetry, yet learning convolutions directly from data has so far proven elusive. Here, we show how initially fully-connected neural networks solving a discrimination task can learn a convolutional structure directly from their inputs, resulting in localised, space-tiling receptive fields that match the filters of a convolutional network trained on the same task. By carefully designing data models for the visual scene, we show that the emergence of this pattern is triggered by the non-Gaussian, higher-order local structure of the inputs, which has long been recognised as the hallmark of natural images. We provide an analytical and numerical characterisation of the pattern-formation mechanism responsible for this phenomenon in a simple model and find an unexpected link between receptive field formation and tensor decomposition of higher-order input correlations.

**Keywords:** Neural networks — convolution — receptive fields — invariance — emergent properties — symmetry

## Introduction

Exploiting invariances in the inputs is crucial for constructing efficient representations and accurate predictions in neural circuits. In neuroscience, translation invariance is at the heart of models of the visual system (DiCarlo et al., 2012; Yamins et al., 2014; Kar and DiCarlo, 2021; Spoerer et al., 2017), while in machine learning, convolutional neural networks are designed to exploit translation invariance (LeCun et al., 1990; Scherer et al., 2010). While the two hallmarks of convolutions, namely localised receptive fields that tile the input space, can be implemented with fully-connected neural networks, learning convolutions directly from inputs in a fully-connected network has so far proven elusive (Urban et al., 2017; d'Ascoli et al., 2019) without elaborate pruning (Pellegrini and Biroli, 2021) or regularisation strategies (Neyshabur, 2020). Whether convolutions can be learnt from scratch has thus been a central problem in neuroscience and machine learning since the seminal work by Olshausen and Field (1996) on unsupervised learning.

Here, we show how initially fully-connected neural networks solving a discrimination task can learn a convolutional structure directly from their inputs, resulting in localised,

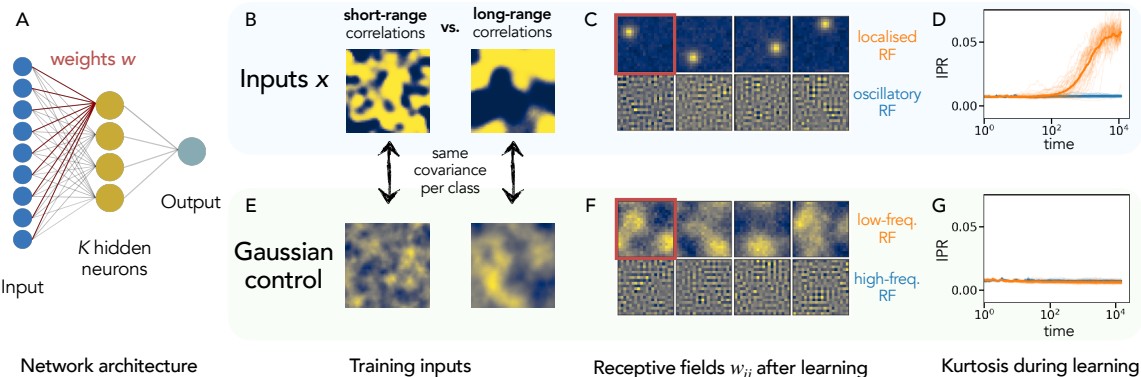

Figure 1: **The emergence of convolutional structure in fully-connected neural networks is driven by higher-order input correlations. A** Two-layer, fully-connected neural network with $K$ neurons in the hidden layer. **B** Networks are trained on inputs drawn from a translation-invariant random process, eq. (1). The task is to discriminate inputs with different correlation lengths. **C** Receptive fields (RF) of some representative neurons taken from a network with $K = 100$ neurons after training. Half the neurons develop localised receptive fields: the magnitude of their weights is significantly different from zero only in a small region of the input space. The other neurons have oscillatory weights. **D** Inverse Participation Ratio (IPR) of each neuron during training. The IPR is large for localised RF, but remains small for oscillatory RF. **E** Gaussian control dataset: the network is trained on a mixture of two Gaussians, each having zero mean and the same covariance as inputs in **B**. **F** Receptive fields after training the network on the Gaussian control data. **G** Inverse participation ratio (IPR), of the receptive fields of a network trained on Gaussian data.

space-tiling receptive fields. By carefully designing data models for the visual scene, we show that this phenomenon relies on the non-Gaussian, higher-order local structure of the inputs, which has long been recognized as the hallmark of natural images (Bell and Sejnowski, 1996). We characterise receptive field formation analytically and numerically, revealing an unexpected link with tensor decomposition of higher-order input cumulants. The receptive fields learnt by the fully-connected networks match the filters found by training a convolutional network on the same task. These results provide a new perspective on the development of low-level feature detectors in various sensory modalities, and pave the way for the study of higher-level invariances in cortical processing.

## Results

**A high-dimensional dataset with tunable higher-order moments** We train two-layer networks (fig. 1A) on a minimal model of natural images capable of capturing higher-order spatial correlations. We generated two-dimensional inputs inputs $\boldsymbol{x} = (x_{ij})$ by first drawing a random vector $\boldsymbol{z} = (z_{ij})$ from a centered Gaussian distribution with a covariance

that renders the input distribution translation invariant along both dimensions. Each pixel in the synthetic image $x_{ij}$ is then computed as

$$x_{ij} = \psi\left(g z_{ij}\right)/Z(g) \tag{1}$$

where $\psi(\cdot)$ is a symmetric, saturating non-linear function such as the error function, $g > 0$ is a gain factor, and the normalisation constant $Z(g)$ ensures that pixels have unit variance for all values of $g$ (see appendix A for details). Intuitively, the gain factor controls the sharpness in the images: a large gain factor results in images with sharp edges and important non-Gaussian statistics (fig. 1B), while images with a small gain factor are close to Gaussians in distribution. The task consists in discriminating inputs with short ($\xi_-$) vs long ($\xi_+$) correlation length. Crucially, inputs have sharp edges, which is a visual indication of higher-order spatial correlations which cannot be captured by a simpler Gaussian model. Indeed, as we show in fig. 1E, samples from a Gaussian distribution with the same covariance as the inputs appear blurry in comparison.

**Learning convolutions directly from stimuli**  We trained two-layer neural networks on this task using vanilla stochastic gradient descent, achieving test accuracy $> 90\%$. We plot the weight vector, or receptive field (RF), of several hidden neurons of the trained networks in fig. 1C. The RF of half of the neurons are *localised*: they only have a few synaptic weights whose magnitude is significantly larger than zero in a small region of input space. Neurons that detect short-range correlations develop different representations, instead converging to highly oscillatory patterns. We can quantify the localisation of receptive fields by computing the Inverse Participation Ratio (IPR) of their weight vector $\boldsymbol{w} = (w_i)$, $\mathrm{IPR}(\boldsymbol{w}) = \left(\sum_{i=1}^{D} w_i^4\right)/\left(\sum_{i=1}^{D} w_i^2\right)^2$. The IPR quantifies the amount of non-zero components of a vector and is commonly used in quantum mechanics and random matrix theory (Metz et al., 2010). We plot the IPR of all neurons during training in fig. 1D. Localised neurons develop a large IPR over the course of training, while the IPR of neurons with oscillatory receptive fields remains very small. Crucially, we found that the RFs of this fully-connected network are spread over the entire input range (fig. S1A) and that they match the filters of a two-layer convolutional network trained on the same task (fig. S1).

**Learning convolutions requires higher-order input cumulants**  As a control, we trained the same networks on a task where inputs for each class are Gaussian with the same covariance as the original data (fig. 1B). These inputs are still translation-invariant, but lack the non-trivial higher-order statistics. Networks trained on these control inputs do *not* form localised receptive fields (fig. 1F), instead converging to oscillatory patterns. The kurtosis of all neurons stays also close to zero throughout learning (fig. 1G). Taken together, these results show that both translation invariance *and* non-trivial higher-order statistics are needed to learn localised receptive fields from scratch.

**Existing theories of learning in neural networks fail to capture the formation of receptive fields**  The dynamics of (deep) linear networks depends only on the input-input and the input-label covariance matrices Saxe et al. (2014) and can therefore not capture the formation of receptive fields, which is driven by non-Gaussian fluctuations in the inputs. Similarly, an analysis of the learning dynamics using the Gaussian Equivalence Theorem (Goldt et al., 2020; Hu and Lu, 2020; Goldt et al., 2021; Mei and Montanari, 2021)

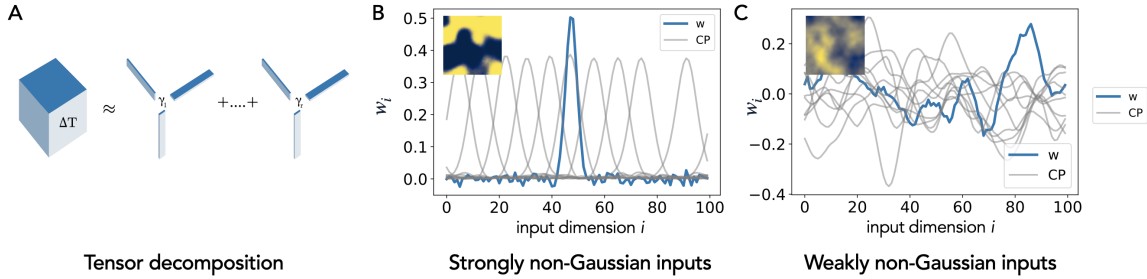

Figure 2: **Non-gaussianity drives pattern-formation in a simplified model of gradient descent dynamics. A** Pictorial illustration of CP decomposition (Kiers, 1998; Kolda and Bader, 2009), a tensor decomposition technique where a tensor (here a three-way tensor) is decomposed into a weighted sum of rank-1 tensors. **B, C** Synaptic weight vectors $\boldsymbol{w}$ (blue) obtained from integrating the GF equation for small (**B**) and large (**C**) values of the gain parameter. The corresponding inputs are shown as insets. In grey, we show the ten leading CP factors $\boldsymbol{u}_k$ of the fourth-order cumulant $\Delta T$ for both datasets, eq. (2). *Parameters*: 1-dimensional inputs, $D = L = 100$, $K = 1$, $\xi^- = 0$, cumulants estimated from a dataset with $P = \alpha D$ inputs, $\alpha = 100$, learning rate $\eta = 0.01$, bias fixed at $b = -1$.

breaks down precisely when localised receptive fields form, highlighting the non-Gaussian nature of their formation. We discuss this issue in more detail in appendix B.

**Connecting receptive fields to data geometry** An analysis of the gradient flow dynamics of a simplified model revealed an interesting connection between receptive fields and data geometry. We studied the learning dynamics of a single neuron with a polynomial activation function. The gradient flow dynamics of this neuron depends only on the covariance matrices $C_{ij}^\mu = \mathbb{E}\left[x_i^\mu x_j^\mu\right]$ and the fourth-order moments $T_{ijk\ell}^\mu = \mathbb{E}\left[x_i^\mu x_j^\mu x_k^\mu x_\ell^\mu\right]$ of each input class $\mu$. Our analysis shows that the 4th-order *cumulant* $\Delta T^\mu$ – obtained by subtracting the Gaussian contribution from $T^{\mu-}$ is crucial for the formation of the RF. A tensor like $\Delta T^\mu$ can be decomposed into its leading *CP factors* (Kolda and Bader, 2009), akin to the eigendecomposition of a matrix,

$$\Delta T = \sum_{k=1}^{r} \gamma_k \boldsymbol{u}_k \otimes \boldsymbol{u}_k \otimes \boldsymbol{u}_k \otimes \boldsymbol{u}_k, \tag{2}$$

where $r$ is the *rank* of the decomposition (see fig. 2A for an illustration of a third-order tensor). When training on strongly non-Gaussian inputs with $\Delta T^\mu \neq 0$, the single neuron develops a localised receptive field which mirrors the localisation of the leading CP factors of $\Delta T^\mu$ (blue and grey lines in fig. 2B). The CP factors also tile the input space. If instead inputs are only weakly non-Gaussian, the CP factors, and hence also the weight, oscillate (fig. 2C).

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

## Appendix A. Data models

We conduct the experiments reported in fig. 1 on a data set that consists of inputs $\boldsymbol{x}$ that can be one- or two-dimensional, divided in $M$ distinct classes. Here, we illustrate the different types of inputs in one dimension.

A data vector of the **non-linear Gaussian process (NLGP)** is given by $\boldsymbol{x}^\mu = Z^{-1}(g)\psi\left(g\boldsymbol{z}^\mu\right)$, where $\boldsymbol{z}^\mu$ is a zero-mean Gaussian vector of length $L$ and covariance matrix

$$C_{ij}^\mu = \left\langle z_i^\mu z_j^\mu \right\rangle = e^{-(|i-j|/\xi^\mu)^2}, \tag{A.1}$$

with $i, j = 1, 2, \ldots, L$. The covariance thus only depends on the distance between sites $i$ and $j$, given by $|i - j|$. The normalisation factor $Z(g)$ is chosen such that $\mathrm{Var}\left(x\right) = 1$. Throughout this work, we took $\psi$ to be a symmetric saturating function $\psi\left(z\right) = \mathrm{erf}\left(z/\sqrt{2}\right)$, for which $Z(g)^2 = 2/\pi \arcsin\left(g^2/(1+g^2)\right)$. We also enforce periodic boundary conditions.

We create the **Gaussian clone (GP)** by drawing inputs from a Gaussian distribution with mean zero and the same covariance as the corresponding NLGP in each class. The covariance of the NLGP can be evaluated analytically for $\psi\left(z\right) = \mathrm{erf}\left(z/\sqrt{2}\right)$ and reads

$$\langle x_i^\mu x_j^\mu \rangle = \frac{2}{\pi Z(g)}\arcsin\left(\frac{g^2}{1+g^2}C_{ij}^\mu\right) \tag{A.2}$$

where we have used that fact that $C_{ii} = 1$. The experiments on Gaussian processes (GP) are thus *not* performed on the Gaussian variables $\boldsymbol{z}$; they are performed on Gaussian random variables with covariance given in eq. (A.2). In this way, we exclude the possibility that the change in the two-point correlation function from applying the non-linearity $\psi$ is responsible for the emergence of receptive fields.

For 1-dimensional inputs, the fact that the covariances of the NLGP and the GP depend only on the distances between pixels $|i - j|$ implies that they are *circulant matrices* (Horn and Johnson, 2012). These matrices display a number of useful properties: they can be diagonalised using discrete Fourier Transform (DFT), and thus any two circulant matrices of the same size can be jointly diagonalised and commute with each other. We use this fact in the analysis of the reduced model to diagonalise the dynamics of the synaptic weights.

We obtain the covariance for 2-dimensional inputs by taking the Kronecker product of the one-dimensional covariance matrix with itself. For any dimension, we indicate the total input size by $D$.

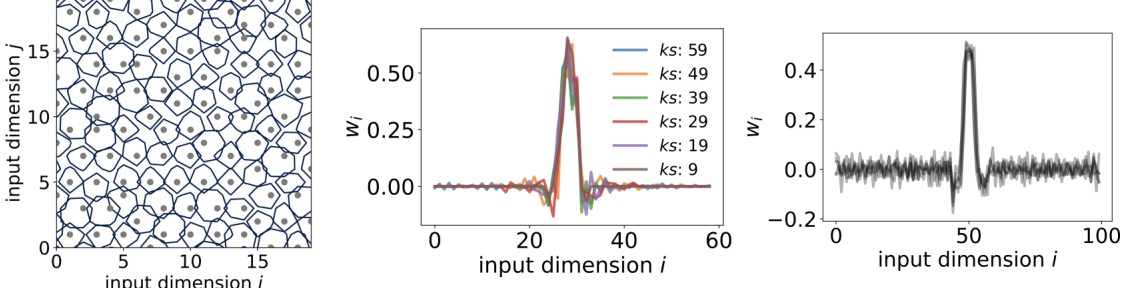

Figure S1: **Receptive fields of fully-connected networks tile input space and resemble the filters learnt by a convolutional neural network**. **A** Centres (grey) and contour lines (blue) of the whole set of localised RF plotted over the 2-dimensional inputs space. Neurons are taken from the network in fig. 1. **B** Overlay of five randomly selected receptive fields from a network trained on a 1D discrimination task, after centering. **C** Filters of a two-layer convolutional network trained on the same task as **B**. Different colours correspond to different kernel sizes $k_S$, ranging from 9 to 59 pixels. *Additional parameters*: gain $g = 3$, batch learning with $P = \alpha D$ inputs, $\alpha = 10^5$, SGD with batch size 1000.

## Appendix B. The limits of Gaussian equivalence in describing the formation of receptive fields

How can we capture the formation of receptive fields theoretically? There exist precise theories for learning in neural networks with linear activation functions Baldi and Hornik (1989); Le Cun et al. (1991); Krogh and Hertz (1992); Saxe et al. (2014, 2019a); Advani et al. (2020). However, the dynamics of even a deep linear network with several layers will only depend on the input-input and the input-label covariance matrices, i.e. the first two moments of the data Saxe et al. (2014). This formalism thus cannot capture the formation of receptive fields, which is driven by non-Gaussian fluctuations in the inputs. An exact theory describing the learning dynamics is available for non-linear two-layer neural networks with large input size $D \to \infty$ and a few neurons $K \sim \mathcal{O}(1)$ in the hidden layer Saad and Solla (1995); Biehl and Schwarze (1995). In this limit, one can derive a set of ordinary differential equations that predict the evolution of the (prediction mean-squared) test error pmse of a network, when training on Gaussian mixture classification Refinetti et al. (2021). In fig. S2, we show the pmse of a network with $K = 8$ neurons trained on the Gaussian control task (blue lines) and verify that this theory yields matching predictions (blue crosses).

This type of analysis has recently been extended from mixtures of Gaussians to more complex input distributions thanks to the phenomenon of "Gaussian equivalence", whereby the performance of a network trained on non-Gaussian inputs is still well captured by an appropriately chosen Gaussian model for the data. This Gaussian equivalence was used successfully to analyse random features Liao and Couillet (2018); Seddik et al. (2019); Mei and Montanari (2021) and neural networks with one or two layers, even when inputs were drawn from pre-trained generative models Goldt et al. (2020); Hu and Lu (2020); Goldt

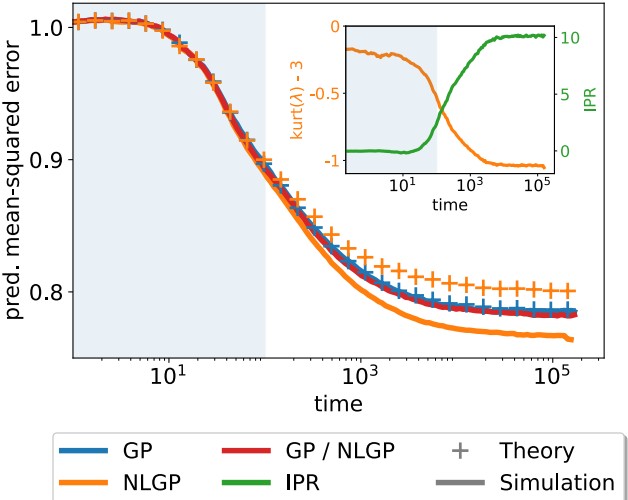

Figure S2: **Existing theories of learning in neural networks break down during the formation of receptive fields.** Prediction mean-squared error of a network with $K = 8$ neurons trained on non-linear Gaussian inputs (NLGP, eq. (1), orange) and on the Gaussian control task (GP, blue) with length scales $\xi^+ = 2\xi^- = 16$. The pmse is calculated using held-out test data during the simulation (solid lines). We also show the test error of the network trained on GP, but evaluated on NLGP data (GP / NLGP, red). The crosses give the pmse obtained from evaluating an analytical expression describing the error of an equivalent Gaussian model (see text). While the analytical expression accurately predicts the error in the beginning of training (blue shaded area), it breaks down for the network trained on NLGP around time $10^2$. This is precisely the time at which the weights start to localise, as measured by the average IPR of the localised weights (inset, green). Simultaneously, the excess kurtosis of the pre-activations of the network decreases (inset, orange). *Additional parameters*: 1-dimensional task with $D = L = 400$, learning rate $\eta = 0.05$. Curves averaged over twenty runs.

et al. (2021); Loureiro et al. (2021). In fig. S2, we plot the test error of a network trained on NLGP data together with the theoretical prediction obtained from applying the Gaussian Equivalence Theorem Goldt et al. (2021) (GET). Initially, the theoretical predictions from the GET (orange crosses) agree with the test error measured in the simulation (orange line), but the theory breaks down around time $\approx 10^2$, when predictions start deviating from simulations.

The breakdown of the Gaussian theory coincides with the localisation of the receptive fields, as measured by their IPR (green line in the inset of fig. S2). The increased localisation of the weights also coincides with a change in the statistics of the pre-activations of the hidden neurons, $\lambda \sim \sum_i w_i x_i$: the excess kurtosis of $\lambda$ (orange line) is initially close to zero,

meaning that $\lambda$ is approximately Gaussian, but decreases as the weights localise, indicating a transition to a non-Gaussian distribution.

We can finally see from fig. S2 that the network is only influenced by the second-order fluctuations in both the NLGP and the GP at the beginning of training, since the pmse for models trained on NLGP and GP initially coincide. Likewise, a network trained on GP and evaluated on NLGP test data has the same test accuracy as the network trained directly on NLGP in the early stages of learning (red line). The higher-order moments of the NLGP inputs start influencing learning only at a later stage, when the IPR of the weight vectors increases and the Gaussian theory breaks down. This sequential learning of increasingly higher-order statistics of the inputs is reminiscent of how neural networks learn increasingly complex functions during training. Simplicity biases of this kind have been analysed in simple models of neural networks Schwarze and Hertz (1992); Saad and Solla (1995); Engel and Van den Broeck (2001); Saxe et al. (2019b); Rahaman et al. (2019) and have been demonstrated in modern convolutional networks Kalimeris et al. (2019). The sequential learning of increasingly higher-order statistics and the ensuing breakdown of the GET to describe learning is a result of independent interest which we will investigate further in future work.

