# OpenReview forum: "Data-driven emergence of convolutional structure in neural networks"
_NeurIPS.cc/2022/Workshop/NeurReps — NeurReps 2022 Poster_

### Official Review · Reviewer_VeSA · 2022-10-09
**Interesting work showing that non-Gaussian inputs are required for emergence of convolutional structure in a fully-connected network.**

**Confidence:** 4
**Soundness:** 3
**Presentation:** 3
**Contribution:** 3
**Overall Rating:** 5

**Summary:**

The authors seek to learn from scratch the structure of convolutions (i.e. localized receptive fields tiling the input space) in a fully-connected (FC) network.

They trained a 2-layer FC network to discriminate between synthetic inputs with short vs. long-range correlations. They found that half of the resulting neurons show localized receptive fields (weights) tiling the input space. When doing the same with inputs that preserve up to 2nd order correlations of the original stimuli, they fond no localized receptive fields, concluding that higher-order statistics drive convolutional structure.

In a second part, they wanted to connect the formation of individual receptive fields with data geometry. They claim that the gradient of the weights of a single neuron is related to the fourth-order moment of the stimulus. They then empirically show that the resulting RF of a neuron “mirrors” the leading factors of a CP tensor decomposition of that fourth-order moment cumulant of the stimulus.

**Questions:**

- What is the performance of the Gaussian control? To isolate the claim of non-gaussianity with that of task performance, I would expect the control network to perform well too (>90%).
- What happens if the number of hidden units is reduced? Should I expect the ratio between neurons with high IPR and low be preserved, or should I expect that localized neurons stays the same.
- What happens with over-complete representations? Would you see more complex kernels (e.g. orientation selectivity)?
- How does the insight relating weight formation and the 4th order cumulant of the inputs extends to complex features beyond blobs or DOGs (e.g. orientation-tuned features)?

**Limitations:**

- The authors could be more transparent about the fact that emergence of convolutional structure from a FC network without regularization and pruning is still an open question under natural stimuli.
- The authors could also point out that their approach works so far for a simple task that requires simple features (blobs or Differences of Gaussians) and no orientation tuning.

**Recommended Decision:**

3: Accept

**Relevance:**

3: Solid fit

**Strengths And Weaknesses:**

The paper shows that higher-order correlations are required to drive convolutional structure using carefully designed stimuli. They do not show this on natural images. The second part of the paper (relating RF formation and data geometry) is less strong and developed, but still promising.

Strengths:

- The writing was clear.
- Great choice of stimuli for tunable high-order correlations! The data model of a non-linear gaussian process (NLGP) yielded a tractable formula of the resulting covariance matrix, useful for the Gaussian process (GP) control.
- The GP control was sensible.
- The paper reveals a gap in theory where existing theories fail to capture the formation dynamics of receptive fields under non-Gaussian inputs.

Weaknesses:

- The paper still does not show if convolutions can arise from a FC network without pruning or regularization tricks when using natural images (which have high-order correlations). It only shows that non-Gaussian structure of inputs is required. This is fine, but the paper's motivation led the reader to believe otherwise.
- The comparison between learnt localized RF features and convolutional features from the CNN was anecdotal and not quantified (Fig S1).
- The authors don’t investigate the –rather surprising– result of two different types of neurons. Fig. 1D shows a strong bimodal distribution of neurons: localized with high IPR vs. high-frequency / “oscillatory”.
- Equation 2 relating dynamics of RF formation and the fourth-order moments of the input was poorly motivated and confusing. How was it derived? Why should we believe it?
- The relationship (described in Figure 2) between a single neuron’s weights and CP factors of delta_T is simply anecdotal. More neurons should be used to make claims. There is also little quantification of what the authors deemed as similar between the shape of the RF and the CP factors.
- Not much was said about the actual convolutional features learnt. In Fig. 1, did the network learn virtually the same feature across neurons at different locations (for high IPR neurons), or were there multiple (different) features?

**Submission Track:**

Extended Abstract (4 Page)

---

### Official Review · Reviewer_orXe · 2022-10-11
**Review of "Data-driven emergence of convolutional structure in neural networks"**

**Confidence:** 4
**Soundness:** 3
**Presentation:** 3
**Contribution:** 3
**Overall Rating:** 6

**Summary:**

This paper shows that training on the task of discriminating short vs. long spatial correlations in generated images with prominent edges can induce convolution-like computation in a fully connected, single hidden layer, nonlinear neural network.

**Questions:**


For the experiment in Figures 1 and S1A, you state “half the [100] neurons develop localised receptive fields” in the caption of Figure 1, but figure S1A seems to show more than ~50 centers. Perhaps this was from another experiment with more neurons?

What happens when the number of neurons is changed: does the ratio of localized vs. oscillatory weights change? If the number of the localized neurons increases, do their receptive fields become smaller, and/or do they overlap more?

The caption of Figure S1 doesn’t exactly match the image: the image does not have plot labels, and what the caption refers to as “B” seems to be the rightmost plot while “C” is the middle plot.

Proofreading:

You are missing parenthetical citations (\citep{}) in the first sentence of the paragraph “Existing theories of learning in neural networks fail to capture the formation
of receptive fields” as well as throughout Appendix B.

“...predict the evolution of the (prediction mean-squared) test error pmse of a network…” -> “...predict the evolution of the prediction mean-squared test error (PMSE) of a network…” (and other instances of PMSE should be capitalized)


**Limitations:**

There is little explicit discussion on implications and limitations of the work, although such discussions are difficult in this short format. Some of my suggestions in this regard were already given above.

**Recommended Decision:**

3: Accept

**Relevance:**

4: Highly relevant

**Strengths And Weaknesses:**

This work is novel, as no other works to my knowledge have shown this phenomenon only through task definition, rather than techniques such as pruning or regularization.

Most claims are well-justified, although further clarification could be added.
* In order to make the claim that “... both translation invariance and non-trivial higher-order statistics are needed to learn localised receptive fields from scratch”, it’s necessary to also test a control task without translation invariance. Also, it may be good to explicitly clarify that these two qualities are not sufficient, particularly as only a specific kind of “non-trivial higher-order statistics” is used experimentally, but this is a very minor point.
* Even given this short format, Figure S1 seems important enough to be in the main paper, particularly as it supports crucial claims made at the end of the paragraph “Learning convolutions directly from stimuli”.
* In the final paragraph of the main paper, “Connecting receptive fields to data geometry”, what exactly is the “analysis” that was performed? I assume it is the theoretical work that generated equation 2, so such non-trivial work should at least be included in an appendix to justify the claims of this paragraph.

Most of the main paper is clear and well-written. However, the final paragraph, “Connecting receptive fields to data geometry”, is the least clear to me. Many variable/constant definitions are missing from equation 2. Are these learning dynamics for a single neuron within a network with other hidden neurons, or with only this hidden neuron? I can only guess from “K = 1” in the caption of Figure 2 that the latter is correct, so this should be clarified in the text.

This work is significant, prompting further investigation of such learning dynamics in artificial and biological contexts. It fits multiple themes of the workshop.



**Submission Track:**

Extended Abstract (4 Page)

---

### Official Review · Reviewer_YeBS · 2022-10-16
**An interesting study showing empirically and theoretically that the emergence of localized receptive fields in a neural network is the result of an interplay between non-gaussian input statistics and the non-linear activation function of neurons.**

**Confidence:** 4
**Soundness:** 4
**Presentation:** 3
**Contribution:** 4
**Overall Rating:** 8

**Summary:**

This abstract shows that the emergence of localized and convolutional receptive fields--in the first layer of a 2-layer network trained on a synthetic dataset of textures--is the result of an interplay between the non-gaussian statistics of the input and the non-linear activation function of the neurons.

**Questions:**

- Why do single neurons develop a localized receptive field which mirrors the localization of the leading CP factors? The interpretation of how the dynamics of equation 2 lead to localized RFs aligned with individual CP modes would be interesting. Are there winner-take-all dynamics taking place in the weight update of equation 2?

- In a neural network with more than one neuron, how does the interplay between neurons lead them to focus on different CP factors?

- How does the present study relate to this other study showing that the non-linearities of the activation functions of neurons can constrain the input eigenmodes combined by single neurons, in the context of the emergence of grid cells:
A unified theory for the computational and mechanistic origins of grid cells
Ben Sorscher, Gabriel C. Mel, Samuel A. Ocko, Lisa Giocomo, Surya Ganguli
https://www.biorxiv.org/content/10.1101/2020.12.29.424583v1.abstract

**Limitations:**

This study gives an interesting theoretical account for the emergence of localized RFs from higher-order moments of the input statistics. But it is unclear to me that it is really explaining the emergence of convolutional structures, e.g. various patterned RFs repeated in space, as suggested by the title: "Data-driven emergence of convolutional structure in neural networks".

**Recommended Decision:**

3: Accept

**Relevance:**

4: Highly relevant

**Strengths And Weaknesses:**

Original and significant work for our theoretical understanding of the formation of localized receptive fields in neuroscience and deep learning.

Quality: the submission seems technically sound to me.

Clarity: the abstract is overall clearly written, but the brevity of the format makes some aspects of the demonstrations difficult to follow. In particular, the terms in equation 2 are not all defined, and the steps to arrive to equation 2 are not explained in the abstract nor the appendix.





**Submission Track:**

Extended Abstract (4 Page)

---

### Decision · Program_Chairs · 2022-10-21

Accept (Poster)